# Surface, Chemical, and Tribological Characterization of an ASTM F-1537 Cobalt Alloy Modified through an Ns-Pulse Laser

**Carlos A. Cuao Moreu** [1] , **Demófilo Maldonado Cortés** [2] , **María del Refugio Lara Banda** [3] ,
**Edgar O. García Sánchez** [1] , **Patricia Zambrano Robledo** [1] and **Marco Antonio L. Hernández Rodríguez** [1,*]

1   Centro de Investigación y Desarrollo Tecnológico (CIDET), Facultad de Ingeniería Mecánica y
    Eléctrica (FIME), Universidad Autónoma de Nuevo León (UANL), Av. Universidad S/N,
    San Nicolás de los Garza 66455, Nuevo León, Mexico; carlos.cuaomr@uanl.edu.mx (C.A.C.M.);
    edgar.garciasc@uanl.edu.mx (E.O.G.S.); patricia.zambranor@uanl.edu.mx (P.Z.R.)
2   Departamento de Ingeniería Mecánica y Electrónica, Universidad de Monterrey, Av. Morones Prieto 4500 Pte.,
    San Pedro Garza García 66238, Nuevo León, Mexico; demofilo.maldonado@udem.edu
3   Centro de Investigación e Innovación en Ingeniería Aeronáutica (CIIIA), FIME-UANL, Aeropuerto
    Internacional del Norte Km 1.7, Carretera a Salinas Victoria, Apodaca 66050, Nuevo León, Mexico;
    maria.laraba@uanl.edu.mx
*   Correspondence: marco.hernandezrd@uanl.edu.mx; Tel.: +52-818-329-4020 (ext. 5774)

**Abstract:** Metallic biomaterials are considered safe materials for the fabrication of orthopedic prostheses due to their mechanical stability. Among this group, cobalt-chromium-molybdenum alloys are commonly used. Nevertheless, adverse reactions on tissues caused by the liberation of metallic ions are a limitation. Therefore, the modification of biometallic material surfaces has become a topic of interest, especially the improvement of the wear resistance to retard the degradation of the surface. In this work, dimples obtained at different processing parameters by an ns-pulse laser were texturized on an ASTM F-1537 cobalt alloy. Surfaces were characterized using scanning electron microscopy (SEM), energy dispersive spectroscopy, and Raman spectroscopy. The mechanical integrity of the surface was evaluated using a 3D surface analyzer and Vickers indentation tests. The tribological response was studied employing a ball-on-disc tribometer under lubricated conditions tracking the coefficient of friction, volume loss, wear rate, and surface damage by SEM. The variation of the laser power, repetition rate, and process repetitions slightly modified the chemistry of the surface (oxides formation). In addition, the rugosity of the zone treated by the laser increased. The texturized samples decreased the wear rate of the surface in comparison with the untreated samples, which was related to the variation of the dimple diameter and dimple depth.

**Keywords:** laser surface texturing; cobalt alloy; tribology; biomaterial

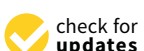

## 1. Introduction

Biomaterials engineering is a discipline that associates medical knowledge and engineering science to increase life expectancy. The fabrication of joint replacement prosthesis is part of this discipline, where metals such as ASTM F-1537 cobalt alloys are commonly used due to their good mechanical properties [1]. However, the damage suffered by a metallic biomaterial in contact with a biological medium causes the deterioration of its physical properties [2]. Likewise, it has been demonstrated that the release of metallic ions inside the human body leads to mutagenic and carcinogenic effects [3].

In the last few years, research in metallic biomaterials has focused on improving their surface functionality through modification processes such as deposition of coatings, passivation-oxide layers, etc. [3]. Laser surface texturing (LST) has excelled in improving the tribological properties of biomedical alloys through reducing friction coefficients and improving load-carrying capacity [4]. Dimples resulting from LST act as lubricant reservoirs and are used as micro-hydrodynamic bearings [5,6]. Moreover, dimples serve as traps for wear debris [7].

Dashtbozorg et al. [8] studied the response of metastable S phase formed on a nitrided AISI 316 L stainless steel, which was modified through nanosecond (ns) Yb-doped fiber laser. Different morphological and chemical changes were observed when the pulse duration of the laser was modified. Cracks were seen near the surface of the 15 ns textured samples, whereas complete loss of the S phase was observed within the 220 ns textured samples. Additionally, the formation of localized high chromium, high nitrogen, low iron, and low nickel in the textured superficial region helped confirm the presence of chromium nitrides. In another study, Menci et al. [9] compared the differences between the textured patterns obtained on a β-Ti alloy using Neodymium-doped yttrium aluminium garnet and Fiber ns lasers. Both conditions can be viable, as the attaching surface in the femoral stem and acetabular cup will stimulate bone growth due to their higher roughness.

Regarding the effect of surface texturing on the tribological properties of biomedical alloys, Zhang et al. [10] developed a petaloid surface texturing on a Co-Cr-Mo alloy artificial joint through laser pulse ablation. Pin on plate tests indicated that petaloid surface texturing is a practical approach to improve the tribological performance of Co-Cr-Mo alloy joint prostheses because of its lubricant reservoir effect. Using tungsten carbide pins, Salguero et al. [11] carried out pin-on-flat reciprocating tests at 5 N over Ti6Al4V texturized samples at different pulse energy densities and scanning speeds, with 1.5 mm diameter under lubricated conditions. A reduction of approximately 80% of wear track volume was obtained for lower scanning speeds (<100 mm/s). The improvement was mainly due to the modification of the alloy by oxidation processes and microstructural changes. Alvarez-Vera et al. evaluated the tribological performance of CoCr microtextured discs against Ultra High Molecular Weight Polyethylene (UHMWPE)cylindrical pins with 3.5 mm diameter, under lubricated sliding conditions using a pin-on-disc tribometer at 10 N. Different laser surface microtextures, including dimple, line, net, and dimple patterns, were selected to generate different hydrodynamic responses. The wear rate and coefficient of friction decreased due to the elastohydrodynamic/hydrodynamic lubrication regime, reducing the wear and surface damage of UHMWPE pins.

Thus, important processing parameters resulting from LST are pulse duration, pulse repetition rate (PRR), maximum laser power, scan speed, process repetitions, etc. By properly selecting the laser parameters, the topography and chemistry of the implant surfaces can be optimized for the desired biomedical application [9]. In addition, the short nanosecond ($10^{-9}$ s (ns)) pulsed laser can induce melt formation and vaporization, which can be exploited differently [9].

However, further work investigating the correlation and influence of the noted processing parameters in the surface of medical-grade CoCrMo alloy is missing from the literature. Therefore, the aim of the present study was to investigate the morphological, chemical, and tribological changes produced by an ns-pulse fiber laser on the surface of an ASTM F-1537 cobalt alloy when the repetition rate, laser power, and process repetitions are variable. Surfaces were characterized using scanning electron microscopy (SEM). The chemical characterization was evaluated via two spectroscopy techniques: energy-dispersive (EDS) and Raman. The tribological response was evaluated via the ball-on-disc transitory wear test under lubricated conditions using optical 3D measurements and SEM.

## 2. Materials and Methods

### 2.1. Materials

Wrought medical-grade CoCrMo alloy ASTM F1537-11 with a chemical composition of 27.8% Cr, 5.5% Mo, 0.31% Ni, 0.25% Si, 0.24% Fe, 0.68% Mn, 0.05% C (wt. %) and balance Co was provided by Carpenter Technology Corporation (Philadelphia, PA, USA). Disks of 30 mm diameter and 4 mm thickness were ground through successive grades of SiC sandpaper up to P1200 grit.

## 2.2. Laser Surface Texturing (LST)

Before the LST procedure, samples were cleaned with acetone in an ultrasonic bath over 15 min. An ns-pulse fiber laser SIF—M2030 (SIDECO, Mexico City, Mexico) was then employed for texturing under the parameters listed in Table 1. As a result, dimples of 200 μm diameter with a separation distance of 200 μm between them were created. Table 2 reports the details of the experimental plan. The employed variables were chosen based on the characteristics that this specific laser system allowed us to modify. The values of laser power and repetition rate were in the range defined by previous authors [12] as optimum. The process repetitions were randomly selected in order to study its effect on texturing. After LST, samples were ground with SiC sandpaper P2400 grit for removing the asperities produced by the laser. Additionally, samples were cleaned in an ultrasonic bath.

**Table 1.** Main characteristics of the employed laser system.

| Laser Specifications | Values |
| --- | --- |
| Maximum laser power [W] | 30 |
| Wavelength [nm] | 1064 |
| Pulse width [ns] | 60 |
| Minimum repetition rate [kHz] | 70 |
| Pulse energy [mJ] | 0.4 |
| Laser beam diameter [μm] | 40 |
| Focusing length of lens [mm] | 281 |
| Scanning speed [mm/s] | 50 |
| Atmosphere | Air |

**Table 2.** Details of the experimental conditions for the laser configuration.

| Employed Variables | Values |
| --- | --- |
| Maximum laser power percentage [%P] | 70; 80; 90 |
| Repetition rate [kHz] | 140; 210 |
| Process repetitions | 3; 6; 10 |

## 2.3. Surface Characterization

SEM observations of the dimple topography were carried out in a Hitachi SU8020 apparatus (Hitachi, Ltd., Tokyo, Japan) with an accelerating voltage of 10 kV.

## 2.4. Chemical Characterization

The elemental distribution of the dimples was obtained using EDS in a JEOL JSM-6510LV SEM apparatus (JEOL Ltd., Tokyo, Japan) operated in backscattering mode, with an accelerating voltage of 20 kV. In addition, oxides formation was analyzed by Raman spectroscopy. The measurements of Raman spectroscopy were recorded using a Thermo Scientific Raman spectrometer. The spectra were collected between 100 and 3500 cm$^{-1}$, using a beam laser (532 nm), with the samples exposed to the air under ambient conditions.

## 2.5. Tribological Tests

The laser texturing conditions of the samples for the tribological tests were selected considering the results of the previous characterization. Surface roughness (Ra) and dimple depth before ball-on-disc tests were determined using a 3D Alicona Edgemaster surface analyzer (Bruker Alicona, Birmingham, UK) (Figure 1). The hardness of the adjacent zone to the textured dimples was measured by means of the Vickers indentation test. The measurements were carried out in a range of (15–30) μm from the dimple (Figure 2). An HMV-2 Shimadzu Micro Hardness Tester (Shimadzu corporation, Kyoto, Japan) under an applied load of 980.7 mN for 15 s was used.

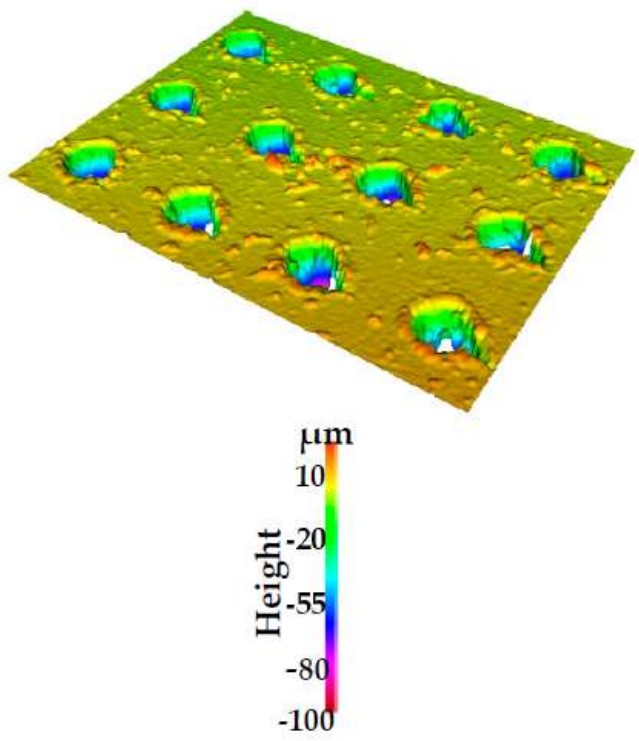

**Figure 1.** Image obtained by the 3D Alicona analyzer on the textured surface.

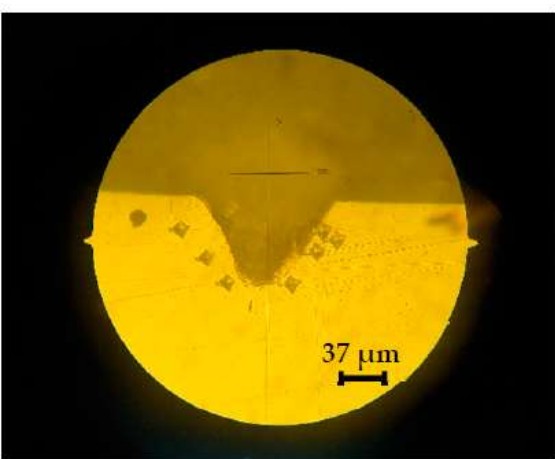

**Figure 2.** Image of the microhardness measurements on the adjacent zone to the texturing.

The tribological performance was evaluated using an STT-1 ball-on-disc tribometer (FIME-UANL, San Nicolás de los Garza, Mexico), testing the textured samples against alumina balls under lubricated sliding conditions. The experimental parameters are shown in Table 3. A circular plane of 2 mm diameter was mechanized on the alumina balls to reach a surface-to-surface contact. The corresponding nominal contact pressure for the surface of the sample was approximately 6 MPa. Simulated body fluid (SBF) was used as lubricant and prepared following the procedure described by Kokubo et al. [13].

**Table 3.** Tribological test parameters.

| Parameter | Value |
|---|---|
| Normal load [N] | 20 |
| Sliding distance [m] | 3166 |
| Sliding speed [rpm] | 210 |
| Alumina ball diameter [mm] | 11 |
| Lubricant | SBF solution |
| Temperature [°C] | 30 |
| Wear track diameter [mm] | 10 |

The coefficient of friction (COF) was continuously recorded during each test. The volume loss was calculated according to the wear track profiles obtained from the 3D surface analyzer. After tribological tests, samples were analyzed using SEM. Two repetitions of tribological tests were performed to verify the reproducibility of the results.

Archard's equation was used to estimate the wear rate of the samples [14]:

$$SN = \frac{V}{k}$$

where $S$ is the relative wear distance (sliding distance), $N$ is the applied load, $k$ is a proportionality constant known as wear coefficient (or wear rate), and $V$ is the volume loss (wear volume).

## 3. Results

### 3.1. Surface Characterization

SEM images of the textured surfaces obtained with the different variables are shown in Figures 3 and 4. A lesser amount of molten material was observed for the dimples synthesized at a repetition rate of 210 kHz compared to 140 kHz. It caused the dimples at 210 kHz to seem more defined. This effect is discussed later. The increase in the laser power also caused an excessive formation of molten material. This formation of molten material could cause heterogeneity during the tribological tests. In the case of the process repetitions, it seemed that three repetitions were enough to produce dimples of considerable depth compared to the other two employed values.

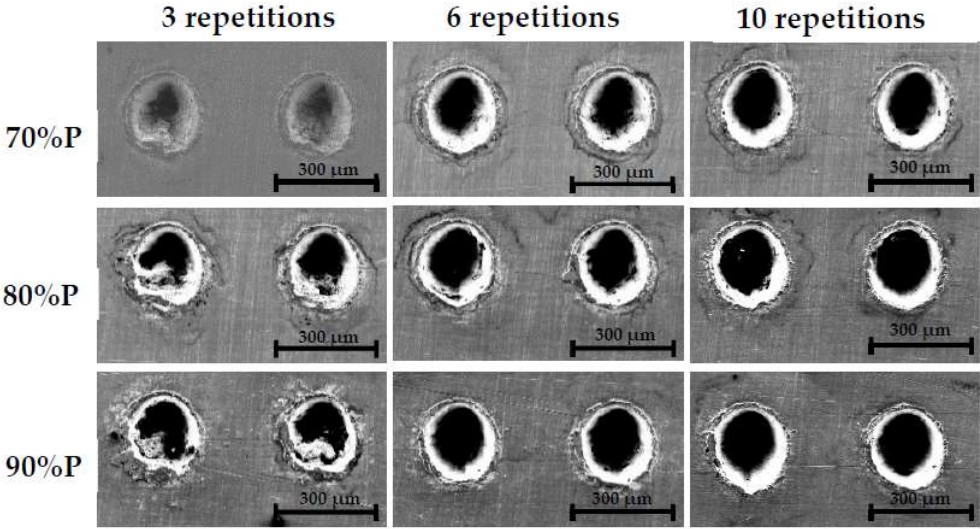

**Figure 3.** SEM images of the dimples obtained on the textured surface of the ASTM F-1537 cobalt alloy employing a repetition rate of 140 kHz.

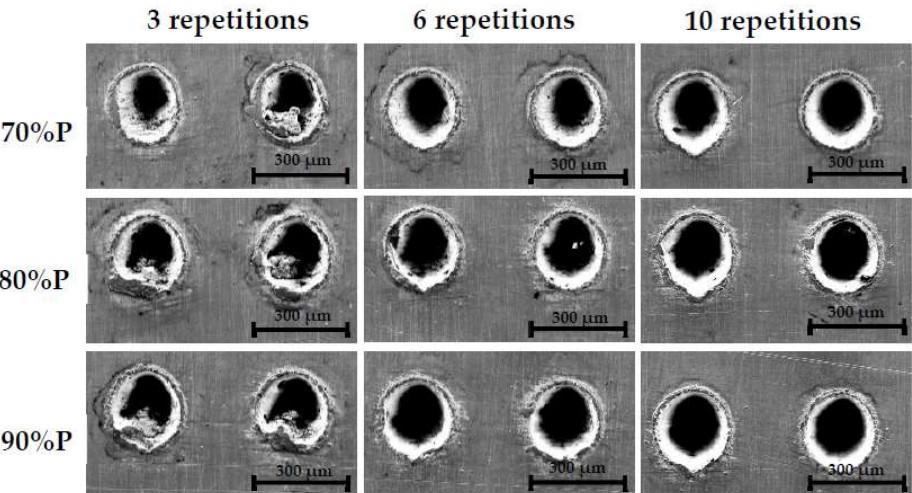

**Figure 4.** SEM images of the dimples obtained on the textured surface of the ASTM F-1537 cobalt alloy employing a repetition rate of 210 kHz.

### 3.2. Chemical Characterization

EDS microanalysis of the textured zone is shown in Figure 5. The oxidation resulting from the texturing process was confirmed by EDS. Raman results (Figure 6) showed a characteristic peak at 850 cm$^{-1}$, which could be associated with Si-O and O-Si-C bonds [15]. In addition, conditions (c) and (e) exhibited a characteristic peak at 665 cm$^{-1}$ that corresponds to CoO and $MoO_3$ phases [16,17].

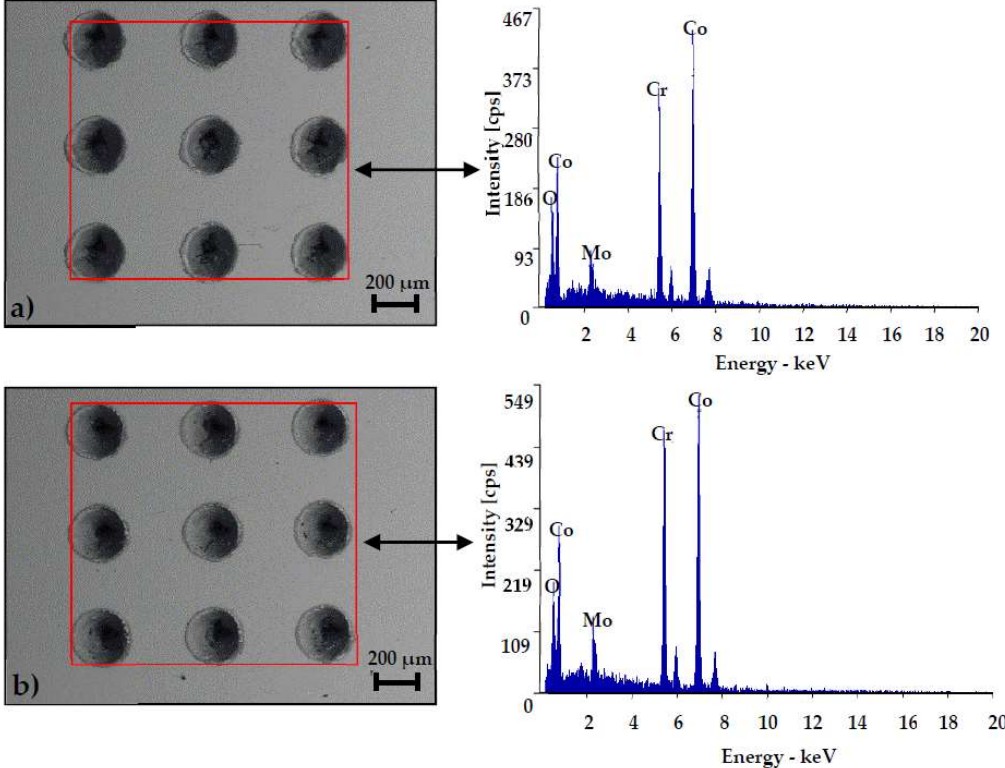

**Figure 5.** EDS elemental mapping obtained by BES-SEM over the dimples of the textured ASTM F-1537 cobalt alloy. The image corresponds to the dimples created with a laser power of 70%, and three times process repetition, at a repetition rate of: (**a**) 140 kHz, and (**b**) 210 kHz.

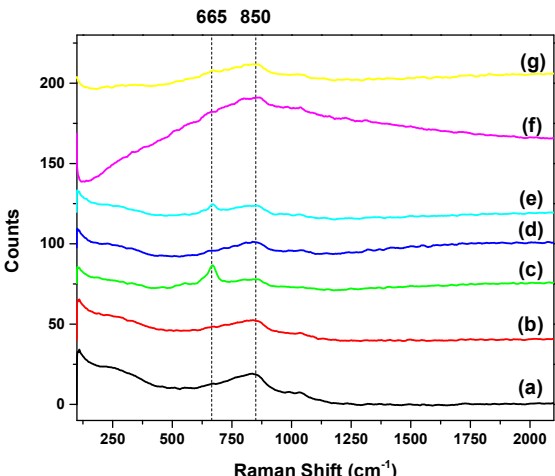

**Figure 6.** Raman spectra of the textured ASTM F-1537 cobalt alloy with six times process repetition at (**a**) non-textured, (**b**) 70%P-140 kHz, (**c**) 80%P-140 kHz, (**d**) 90%P-140 kHz, (**e**) 70%P-210 kHz, (**f**) 80%P-210 kHz, and (**g**) 90%P-210 kHz.

### 3.3. Tribological Tests

The repetition rate was set at 210 kHz because this value allowed more defined dimples. There was no significant difference in the elemental composition between the dimples obtained at the two employed repetition rates. Therefore, the value range of the other variables was reduced, as is shown in Table 4. The 90% laser power was removed because of the excessive formation of melted material around the dimples. The process repetitions were reduced considering that at three repetitions, a considerable depth could be achieved. An additional dimple diameter value was employed according to the reported by previous authors about its influence on the tribological performance of the surface [18].

**Table 4.** Fixed experimental conditions for the laser configuration.

| Employed Variables | Values |
|---|---|
| Repetition rate [kHz] | 210 |
| Maximum laser power percentage [%P] | 70; 80 |
| Process repetitions | 3; 4 |
| Dimples diameter [μm] | 100; 200 |

Notably, the percentage of density was set at 9%. This term refers to the area that is covered by the microcavities concerning the total surface area of contact [19], and it is calculated according to the following equation [20]:

$$N = \frac{\left[\left(\pi D^2\right) \times \rho\right]}{\pi d^2}$$

where $N$ is the total number of dimples, $D$ is the diameter of the disk sample, $\rho$ is the percentage of density, and $d$ is the diameter of the circular dimple. It has been reported that a percentage of density between 7 and 20% increases more entrapment possibility of wear debris and hence reduction in wear [20]. The nomenclature of the employed samples for the tribological tests is exhibited in Table 5, and it is based on the fixed experimental conditions shown in Table 4.

**Table 5.** Description of the employed samples.

| Sample Name | Description |
|---|---|
| M0 | Untreated |
| M1 | 100 μm diameter, 70%P, 3 repetitions |
| M2 | 100 μm diameter, 70%P, 4 repetitions |
| M3 | 100 μm diameter, 80%P, 3 repetitions |
| M4 | 100 μm diameter, 80%P, 4 repetitions |
| M5 | 200 μm diameter, 70%P, 3 repetitions |
| M6 | 200 μm diameter, 70%P, 4 repetitions |
| M7 | 200 μm diameter, 80%P, 3 repetitions |
| M8 | 200 μm diameter, 80%P, 4 repetitions |

Surface roughness (Ra) and dimples depth ($\Delta Z$) results are shown in Table 6. Microhardness results are shown in Table 7. The texturing process increased the Ra of the surface. The different experimental conditions had a remarkable effect on the values of dimple depth. This effect is discussed later. The microhardness of the adjacent zone to the dimples decreased compared to the microhardness of the untreated sample (M0). This could be associated with the heat produced by the texturing.

**Table 6.** Surface roughness and dimple depth of the employed samples.

| Sample | Ra (μm) | $\Delta Z$ (μm) |
|---|---|---|
| M0 | $0.47 \pm 5$ | - |
| M1 | $1.25 \pm 3$ | $53 \pm 9$ |
| M2 | $4.5 \pm 7$ | $85 \pm 10$ |
| M3 | $2.16 \pm 6$ | $60 \pm 8$ |
| M4 | $2.8 \pm 5$ | $90 \pm 5$ |
| M5 | $2.73 \pm 4$ | $193 \pm 11$ |
| M6 | $1.78 \pm 5$ | $224 \pm 10$ |
| M7 | $2.00 \pm 4$ | $292 \pm 8$ |
| M8 | $1.35 \pm 6$ | $354 \pm 10$ |
| Alumina balls | $0.14 \pm 2$ | - |

**Table 7.** Vickers hardness of the adjacent zone to the textured dimples.

| Sample | HV |
|---|---|
| M0 | $466 \pm 18$ |
| M2 | $387 \pm 20$ |
| M3 | $393 \pm 7$ |
| M6 | $409 \pm 8$ |
| M7 | $355 \pm 20$ |
| M8 | $348 \pm 30$ |

Wear rate results are shown in Figure 7. COF values are shown in Figure 8. The texturing process decreased the wear rate of the surface. The differences between the wear rates are discussed later. The COF did not fluctuate considerably. In addition, for almost all the texturing conditions, the COF decreased.

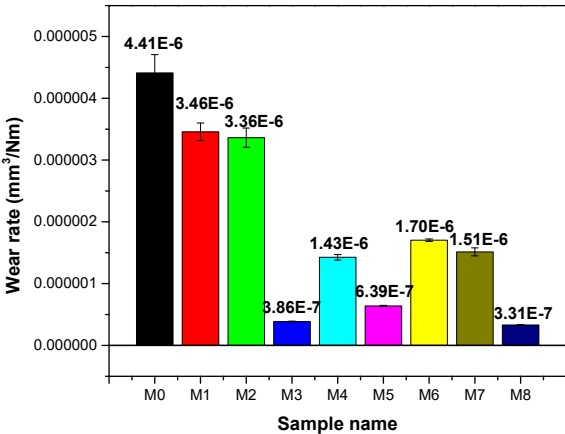

**Figure 7.** Wear rates calculated for each one of the employed samples.

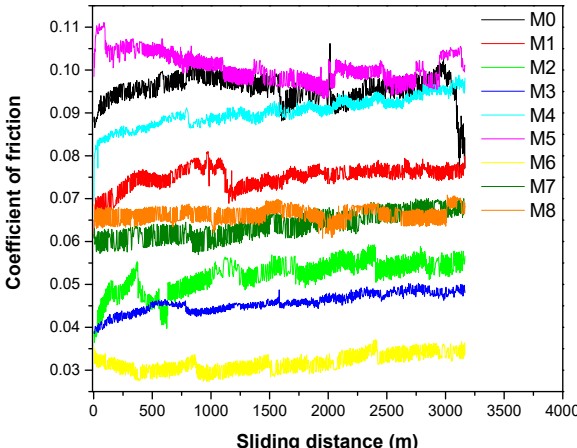

**Figure 8.** Variation of the coefficient of friction obtained at the surface of the employed samples.

Surface morphologies obtained by SEM on the wear tracks of the untreated sample are shown in Figure 9. Abrasion (Figure 9a) and adhesion (Figure 9b) wear mechanisms could be identified. The morphology of the wear track for the textured samples (Figure 10) showed that the dimples decreased the severity of wear compared to the untreated sample. The differences among the worn surfaces of the textured samples are discussed in the next section.

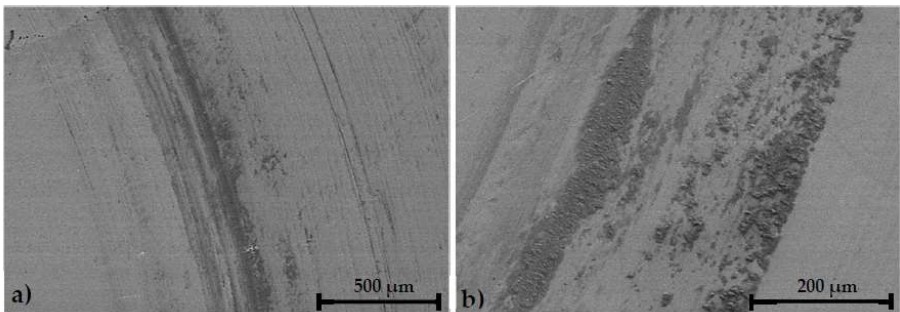

**Figure 9.** SEM micrographs obtained after the tribological test on the wear track of the M0 sample: (**a**) abrasion, and (**b**) adhesion wear mechanisms.

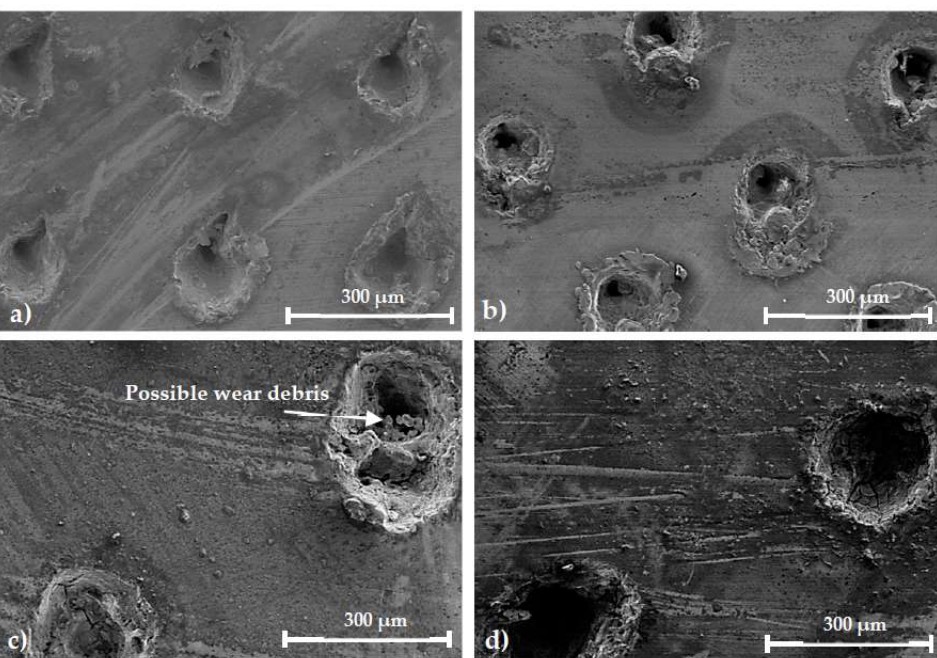

**Figure 10.** SEM micrographs obtained after the tribological test on the wear track of the samples: (**a**) M2, (**b**) M3, (**c**) M6, and (**d**) M8.

## 4. Discussion

Dimples with a repetition rate of 140 kHz showed molten material on their edge and center, which caused the dimples to be irregular (Figure 3). Further, dimples created with a repetition rate of 210 kHz reached a more defined circle (Figure 4). This behavior may be associated with the "plume shielding" effect noted by Allahyari et al. [21], related to the high repetition rates. Plume shielding occurs in ablative conditions where the laser pulses interact with the ablation cloud, inducing absorption and scattering and reducing the material removal rate.

Additionally, the laser power increase produced higher pulse energy, increasing the amount of molten material on the edges of the dimple, especially on the dimples obtained at 90%P. This large formation of material could negatively affect the tribological performance of the surface by serving as third bodies that could accelerate wear. Notably, an apparent increase in the depth of the dimples could be seen at six and ten repetitions compared to three repetitions.

It could be noticed in the EDS analysis that the intensity of O is similar for each one of the employed frequency values (Figure 5). Thus, the variation in frequency did not have a relevant impact on the oxidation resulting from the surface texturing. The Si bonds identified in the Raman spectra (Figure 6) could be associated with the grinding process with SiC sandpaper. The metallic oxides (CoO and $MoO_3$) resulted from the oxidation during laser texturing. However, Raman spectroscopy did not have the needed sensitivity for identifying more oxides.

The LST increased the roughness of the surface (Table 6). This may be related to the formation of molten material around the dimples. A wide range of dimple depth was obtained. Higher laser power and process repetitions increased the depth of the dimples. In addition, a wider dimple diameter increased the depth of the dimples. This effect may be attributed to better use of the laser energy with larger dimples. The LST decreased the hardness of the zone adjacent to the dimples slightly (Table 7). This phenomenon might be due to this zone became a heat-affected zone because of the heat transfer from the dimple. It should be clarified that it was difficult to measure the Vickers indentations on the closest zone to the dimples. It could be expected that the hardness of this zone should have increased because of the rapid solidification of the molten material [22,23].

The surface texturing as surface treatment decreased the wear rate of the surface compared to the untreated material (Figure 7). However, remarkable variations among the wear rates of the different textured samples were noticeable. The wear rates of M1 and M2 samples were the highest compared with the M3 and M4 samples, which were in the group with the lowest wear rates. Notably, the unique difference between M1–M2 and M3–M4 samples was the laser power percentage (Table 5); the influence of the dimple size in the improvement of the surface tribological properties was clearly seen. According to Figure 4, a wider internal diameter of dimple was obtained with higher laser power. This effect influenced the wear resistance of the textured surface. These results support the theory put forward by Schneider et al. [24] that the dimple size plays an essential role in trapping wear debris, storing lubricant, and in the hydrodynamic pressure built by the lubricant film. In the case of the M5–M8 samples, their wear rates were relatively low. This may be due to the dimple size effect noted previously, considering that dimple size in this set of samples was even bigger. Nevertheless, apparent differences among the wear rates of M5–M8 samples could be seen. The convergence of the dimple size with another critical variable, dimple depth, could have influenced the behavior of the wear rates. In agreement with the results reported by Ji et al. [25], the dimple depth has a critical impact on the pressure distribution applied by the lubricant film. Further, no single textured condition exhibits the highest wear resistance; their optimization depends on the employed tribological parameters [24]. According to the selected tribological parameters, the M8 sample showed the lowest wear rate in the present study.

In general, the behavior of the COF was stable (Figure 8). This could be associated with the elastohydrodynamic lubrication between both contact surfaces. The results were consistent with Schneider et al. [24], who reported similar COF values in a metal (texturized)–alumina contact under lubrication. However, texturing effect in friction was evident because the magnitude of the COF for most textured samples was lower than for the untreated sample. The textured patterns increase the lubricant film between the two surfaces because they serve as lubricant reservoirs [22]. Importantly, the lubrication regimen could be considered elastohydrodynamic because the two surfaces are not totally conforming [26]. Further, not all texturing conditions reduced COF. For example, the COF of M4 and M5 samples kept similar values to the COF of the M1 sample. This phenomenon might be due to the increase in rugosity for the M4 and M5 samples (Table 6) after texturing that increased the number of asperities and raised friction between surfaces [27]. This influence of the rugosity in the COF could also suggest a mixed lubrication because the sliding speed was not excessively high [28]. Even though laser texturing increased the rugosity of surfaces, the COF behavior of all samples was nearly constant during the test. Additionally, the sample with the lowest COF did not exhibit the lowest wear rate also. The M2 sample showed one of the lowest COFs, but its wear rate was one of the highest. Moreover, the M8 sample showed the lowest wear rate, and its COF was not the lowest. However, the relation between wear rate and COF was not always inversely proportional for all texturing conditions; the M3 sample exhibited a lower wear rate and a lower COF. It can be inferred that the M3 sample was the condition with the best tribological performance because the two aforementioned variables were among the lowest. The COF does not always keep a directly proportional relationship with the wear rate. The COF is considered a system property as its behavior depends on the employed conditions, whereas the wear rate is a material property [29].

Wear mechanism of abrasion could be seen, as evidenced by grooves along the motion direction (Figure 9a). In addition, adhered material could be identified on the wear track (Figure 9b). This adhesion of material could be related to particles pulled out from the surface due to wear. These resulting particles could have helped to form three-body abrasion, and after been deformed by the cyclic sliding, they adhered on the wear track. A less severe abrasion mechanism could be distinguished on the textured samples (Figure 10). This may be because the dimples reduced the real contact area between the surfaces. Further, the increase in the hydrodynamic lubrication improved the load-carrying capacity of the

surface, which in turn decreased the wear according to some authors [23,30]. Additionally, a certain deformation on the edge of the dimples could be identified (Figure 10a,b). This phenomenon might be due to the increase in the pressure of contact due to the decrease in the contact area, which deformed plastically the edges before the dimples after some load cycles. No evidence of adhesion could be observed, possibly because the dimples serve as traps for the wear debris (Figure 10c). These particles within the dimples are a mechanism to prevent surface wear because there is no interaction between the debris and the sliding surfaces.

## 5. Conclusions

The parameters of LST studied in the present work considerably affected the morphology of the obtained dimples. The repetition rate is related to the ability of the laser to create defined patterns. The laser power and the process repetition influenced the dimple depth and the amount of molten material formed around the dimples. This metallic molten material ($CoO$ and $MoO_3$) increased the surface roughness. The increase in the dimple diameter from 100 μm to 200 μm decreased the wear rate of the surface, and the interaction between the dimple diameter and the dimple depth caused variations in the wear rate of the samples. The sample M3 exhibited the best tribological performance, and its wear rate was 10 times lower than the wear rate of the untreated sample. The COF was relatively stable during all the tests, and the LST decreased the friction for most of the selected parameters over a range from 26 to 68%. In addition, the LST improved the response of the surface to the severe abrasion and adhesion wear mechanisms.

**Author Contributions:** Conceptualization, C.A.C.M., D.M.C., and M.A.L.H.R.; methodology, C.A.C.M.; validation, P.Z.R. and M.A.L.H.R.; formal analysis, C.A.C.M. and M.A.L.H.R.; investigation, C.A.C.M. and D.M.C.; resources, M.d.R.L.B., E.O.G.S., P.Z.R., and D.M.C.; data curation, D.M.C. and E.O.G.S.; writing—original draft preparation, review, and editing, C.A.C.M. and M.A.L.H.R.; supervision and project administration, M.A.L.H.R. and D.M.C. All authors have read and agreed to the published version of the manuscript.

**Funding:** This research received no external funding.

**Institutional Review Board Statement:** Not applicable.

**Informed Consent Statement:** Not applicable.

**Data Availability Statement:** Data presented in this article are available upon request from the corresponding author.

**Acknowledgments:** The authors wish to thank to "Consejo Nacional de Ciencia y Tecnología" CONACYT for providing a doctoral research fellowship, and to the research grant from UANL (CB 239808 STT-Tribometer). The also express their gratitude to CIDET (FIME), CIIIA, CIIDIT, and UDEM for allowing them to perform each test.

**Conflicts of Interest:** The authors declare no conflict of interest.

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
