# Peer review of "Surface, Chemical, and Tribological Characterization of an ASTM F-1537 Cobalt Alloy Modified through an Ns-Pulse Laser"

_metals, doi:10.3390/met11111719_

Round 1
Reviewer 1 Report
metals-1430400
Title: Surface, chemical, and tribological characterization of an ASTM F-1537 cobalt alloy modified through ns-pulse laser
The submitted paper discussion the surface, chemical, and tribological characterization ASTM F-1537 cobalt alloy modified through ns-pulse laser. The proposed manuscript within the scope of the Metals journal. Some of the results observations are interesting, however, this manuscript can’t be accepted for publication in this status, if the paper can be improved in the following areas, it would add more value to the readers:
- The abstract should contain results of the original works and should be summarized well.
- For the experiments: authors present laser treatment conditions. The parameter selection should be systemically well described including laser power percentage, repetition rate, and process repetitions.
- Compared with the characteristics result of the sample, the original and after laser modification characteristics should be summarized clearly.
Reviewer 2 Report
The work is devoted to the current direction of engineering the surface of a biocompatible material. Such work has been actively carried out in the past three decades, which was due to the development of laser technology and an increase in its availability.
The main problem of this work is the carelessness in the presentation and description of the results.
1. Introduction.
The literature review is based on modern sources.
Comment:
A laconic and specific goal of the work has not been formulated. Instead, a description of the work performed is provided.
2. Materials and methods.
The section contains a description of the material and research methods.
Notes:
Explain in more detail the method for determining the hardness. How close to the fossa did you do the indentation?
Table 3. Give the explanation of the abbreviation SBF.
3. Results.
The results are shown casually. The section needs to be completely revised so that the reader understands what has been done and what has been received.
Here are just a few notes on this section:
Figure 3. Edit the description so that you can accurately separate the 4 images shown in this figure.
Based on the presented results, it is completely unclear why the authors chose for tribological tests the samples obtained with the parameters indicated in Table 4. The brief description in lines 229-237 is not supported by adequate explanations of the results in Sections 3.1-3.2.
Table 5. It is not clear from the results where these samples came from.
Figures 5 and 6. It is necessary to separate figures from tables.
Table in Figure 6. Data on the hardness of the sample M8 are not given.
The description of the results of tribological tests gives absolutely no understanding of what the authors received as a result of friction. There is no description of graphs, no description of the morphology of worn surfaces. For some reason, Figure 9 shows markers a) and b).
4. Discussion.
This section describes the main findings and contains a discussion of causation.
As a side note:
The statement about the elastohydrodynamic effect is greatly exaggerated. At a sliding speed of 0.1 m / s, this effect is weak. The friction regime in this case is mixed.
5. Conclusions.
Generally consistent with the results when analyzed in conjunction with the above discussion.
Round 2
Reviewer 1 Report
Authors has been revise and response reviewer's comments. I suggestion this manuscript can be accepted for publication in this status.
Reviewer 2 Report
The authors took into account the comments and made the necessary corrections to the manuscript.
I suppose that the article in its current form can be accepted for publication.